# Complexity in Implementing Community Drowning Reduction Programs in Southern Bangladesh: A Process Evaluation Protocol

**DOI:** 10.3390/ijerph16060968

**Published:** 2019-03-18

**Authors:** Medhavi Gupta, Aminur Rahman, Kamran ul Baset, Rebecca Ivers, Anthony B. Zwi, Shafkat Hossain, Fazlur Rahman, Jagnoor Jagnoor

**Affiliations:** 1The George Institute for Global Health, University of New South Wales, Level 5/1 King St, Newtown, NSW 2042, Australia; 2Centre for Injury Prevention and Research, Bangladesh, House 162B, Road 23, New DOHS, Mohakhali, Dhaka 1206, Bangladesh; aminur@ciprb.org (A.R.); kamran@ciprb.org (K.u.B.); shafkat@ciprb.org (S.H.); fazlur@ciprb.org (F.R.); 3School of Public Health and Medicine, Faculty of Medicine, UNSW Australia, Wallace Wurth Building, Botany Street, Kensington, NSW 2052, Australia; rebecca.ivers@unsw.edu.au; 4School of Social Sciences, Faculty of Arts and Social Sciences, UNSW Australia, Morven Brown Building, Kensington, NSW 2052, Australia; a.zwi@unsw.edu.au; 5The George Institute for Global Health, University of New South Wales, 311-312, Third Floor, Elegance Tower, Plot No. 8, Jasola District Centre, New Delhi 110025, India; jjagnoor1@georgeinstitute.org.in

**Keywords:** drowning, process evaluation, evaluation studies, community health workers, rural population, child, education, Bangladesh, injury

## Abstract

Living and geographical conditions in Bangladesh expose children to a high risk of drowning. Two programs operating in the Barishal Division of Bangladesh aim to reduce drowning risk through the provision of crèches (Anchal) and swim and rescue classes (SwimSafe). Anchal provides a safe environment with early childhood education to children aged 1–5 years old, while SwimSafe teaches children aged 6–10 years old basic swimming and rescue skills. Despite evidence for their effectiveness, it is unclear under which conditions these programs best operate. This protocol describes a project that seeks to conduct a process evaluation and gender analysis to identify implementation inefficiencies and contextual considerations for improved sustainability of the programs. A mixed- method approach using both qualitative and quantitative data will be used. Quantitative program data will be analysed to measure program utilisation, delivery and reach, while qualitative data will be collected via key informant in-depth interviews (IDIs), focus group discussions (FGDs) and observations. The process evaluation of the Anchal and SwimSafe programs provides an opportunity for implementers to identify practical strategies to improve program delivery and improve contextual adaptability of these programs. Furthermore, the findings may provide guidance to other implementers aiming to deliver community-based programs in rural lower-middle income contexts.

## 1. Introduction

Globally, children aged 1–14 years in lower-middle-income countries (LMICs) are at the greatest risk of morbidity and mortality from drowning. This burden is particularly significant in Bangladesh, where 43% of deaths in children aged 1–4 years occur from drowning [1,2]. The rate of fatal drowning in Bangladesh is 122 per 100,000 children in 1–4 years old and 23 per 100,000 in children aged 5–9 years old, significantly higher than the average of 16 per 100,000 in the general population. Non-fatal drowning rates show a similar trend, with a rate of 3058 per 100,000 children in 1–4 years old children and 466 per 100,000 in children aged 5–9 years old, compared to an average rate of 318 per 100,000 in the population [1,2]. Environmental conditions in Bangladesh substantially increase the risk of drowning for children, as much of the country is prone to frequent flooding and waterlogging [3,4]. Many children live close to open water sources such as ponds, rivers and beaches due to high population density coupled with economic dependence on fishing industries [5].

The World Health Organization has recommended the implementation of four community-based interventions that reduce drowning in rural LMIC contexts in response to the United Nation’s Sustainable Development Goal 3 to reduce preventable deaths of children under the age of 5 years to 25 per 1000 live births [1,6]. Currently, the under-5 mortality rate in Bangladesh is 34.2 per 1000 live births [7]. Key drowning interventions for young children include the installation of barriers to control access to water (such as playpens and fencing), the provision of safe spaces away from water for pre-school aged children with capable child care, teaching school-aged children basic swimming and rescue skills and training adult bystanders in safe rescue and resuscitation [1,8].

In 2016, the Centre for Injury Prevention and Research, Bangladesh (CIPRB) developed the Project *BHASA* drowning reduction program based on these identified effective interventions in an effort to reduce the burden of drowning in Bangladesh. Two of these effective interventions within Project *BHASA* are Anchal and SwimSafe [9]. The Anchal program provides community-based crèches where children aged 1–5 years are cared for within a supervised, protected environment with early childhood development. The SwimSafe program provides survival/basic swimming skills and rescue techniques training to children aged 6–10 years. These reflect WHO recommendations for the provision of safe spaces for pre-school aged children and swim training for school-aged children respectively. 

## 2. Program Details

There are three Project *BHASA* intervention *upazilas* (sub-districts) in the Barishal division: Kalapara, Taltoli and Betagi *upazilas*. Evaluation will be conducted across all intervention sites. *Upazilas* are the lowest level of administrative units in Bangladesh, and typically have a population between 250,000 to 400,000 people. 

### 2.1. Program A—Anchal Program

There are 400 Anchal centres under the program across all three intervention *upazilas*. Each Anchal Centre is managed by a trained Anchal Maa (caregiver of a crèche) and Anchal Assistant (caregiver’s assistant) who supervise 20–25 children from 9 a.m. to 1 p.m. six days a week. During this time, children are involved in early childhood development (ECD) activities aimed at stimulating children’s physical, intellectual, linguistic, social and emotion development. Children aged 1 to 5 years old are eligible to attend. Anchal centres are held in a suitable room in the Anchal Maa’s home, which is equipped with educational materials for child stimulation and barriers to create an enclosed space [10]. Table 1 lists the target number of centres that CIPRB aimed to implement for each *upazila*. Figure 1 below presents the logic model for Anchal, describing key inputs, outputs and assumptions for this program. 

### 2.2. Program B—SwimSafe Program

The SwimSafe program provides children aged 6–10 years of age with a 21-step swimming course delivered in 12 sessions aimed at teaching basic swimming and rescue skills. The program was developed in collaboration with experts from The Alliance for Safe Children, UNICEF and Royal Life Saving Society Australia (RLSSA), catering for contextual requirements [11]. Each session runs for an hour every day until course completion. Classes are provided by 103 trained Community Swimming Instructors (CSIs) in 65 modified ponds. These are local ponds that have been specifically identified and modified with bamboo platforms upon which swim classes can be conducted safely. Children attend the classes until they have reached the required competencies, including swimming 25 m unaided, floating in water for 30 seconds, and demonstration of rescue techniques. Each class is taught in groups of five by one CSI while remaining children watch from outside the pond. SwimSafe classes are held during the monsoon months from mid-June to mid-November [12]. Table 1 lists the target number of centres for each *upazila*. Figure 2 below presents the logic model for SwimSafe, describing key inputs, outputs and assumptions for this program. 

### 2.3. Monitoring Structures

Both Anchal and SwimSafe programs are managed at the field level by three levels of staff. Firstly, a total of 20 Supervisors directly manage CSIs and Anchal Staff within their communities. Each Supervisor manages 25–35 Anchal centres or 10–15 ponds and receives a one-day orientation to the program on joining. The ECD component of Anchal is additionally monitored by Anchal Monitoring Officers (AMOs) and an ECD specialist. An Area Coordinator oversees the programs’ operations in each *upazila*, managed by an overarching Program Coordinator. Program trainers are also employed to provide initial and ongoing training to Anchal staff and CSIs. All AMOs and three of the supervisors are female.

CIPRB has also founded Village Injury Prevention Committees (VIPC) in each community. These committees are comprised of local formal and informal leaders. VIPC committees facilitate interactions and engagement between CIPRB and members of the community. VIPC members also oversee the programs’ implementation within the community and participate in recruitment of Anchal Maas and CSIs. They are also tasked with raising awareness on drowning prevention in their communities. Committees meet monthly to discuss these issues. 

Although both Anchal and SwimSafe have been found to be cost-effective in reducing drowning within communities [9] analysis of the many components of these complex programs and how they contribute to the programs’ success has yet to occur. These include investigations of supervision structures, fidelity to program standard operating procedures (SOPs) and monitoring procedures. A process evaluation is thus vital for identifying whether implementation has occurred as intended, and to identify inadequacies and opportunities for improvement in the programs’ delivery [13,14]. A process evaluation for complex interventions can help explain for whom, how and why the intervention had a particular impact. Such evaluations address the question ‘Is this intervention acceptable, effective and feasible for this population?’ Gaining a clear understanding of the causal mechanisms of complex interventions is vital in being able to sustain, scale up or deliver an effective intervention in other settings [15]. Therefore, this protocol outlines the frameworks and methodologies that will be used in the process evaluation of Anchal and SwimSafe. 

## 3. Materials and Methods

### 3.1. Theoretical Approach and Frameworks

Realist theory posits that programs may have differing effects across contexts and participants. To understand this dynamic, the underlying mechanisms behind the effect of a program requires examination [16,17]. Underpinned by this realist approach, the current process evaluation will be adapted from the United Kingdom’s Medical Research Council [15]. Data collection and analysis will seek to identify the ‘true’ mechanisms and processes that shape the current status quo, as influenced by the external social and cultural context [18,19].

In addition, comprehensive process evaluations that seek to understand unintended consequences, such as on gender norms and behaviours, are better able to adapt to contexts and ensure maximal effectiveness, while also considering the equity of outcomes [20,21]. Gender analysis will be conducted based on the Gender Integration Framework to comprehensively identify the effect of the programs on gender roles, perceptions and behaviours [22]. The following diagram displays how these frameworks interact to provide a holistic picture. Figure 3 presents a conceptual framework for how the above two frameworks will be brought together develop a comprehensive picture of the programs’ implementation success. 

### 3.2. Data Collection

A mixed- method approach using both quantitative and qualitative data will be used to triangulate the program monitoring data with key informant in-depth interviews (IDIs), focus group discussions (FGDs) and program observations. 

#### 3.2.1. Quantitative Data

As part of the enrolment process for the programs, staff visit all households in the community and conduct a baseline survey to gather information on the community’s demographics. Program monitoring data are also collected, including participant demographics, participant attendance, graduation rates, dropout rates and standardised monitoring forms completed by supervisors. This data is collected and managed using REDCap electronic data capture tools through hand held tablets.

Quantitative data from across the Anchal and SwimSafe programs will be analysed to gather metrics such as child attendance and dropout rates, staff retention and age and gender differences in enrolment. Table 2 outlines these instruments and the quantitative information available from each.

#### 3.2.2. Qualitative Data

The methodology used will be guided by the COREQ guidelines for qualitative research [23]. Qualitative data will be collected by trained data collectors through observation of program delivery and processes, in-depth interviews (IDIs) and focus group discussions (FGDs) in each of the *upazilas*. IDIs will provide an understanding of individual-level responses to the programs, while FGDs will provide insights to wider community perspectives and cultural norms [24]. Observations provide an opportunity to identify how the program delivery is impacted by context, and captured data on community staff and participant behaviour that is difficult to describe verbally. All data collection will occur face-to-face, unless a participant is not available in person in which case the interview will be conducted over the phone. To build a picture of program provision success and challenges, purposive sampling will be used to select individuals with the most insight to program operations such as community-level staff (Anchal Maas, Anchal Assistants and CSIs), program implementing staff (Supervisors, Area Coordinators, trainers and AMOs) and headquarters staff. To ensure that representative end-user experiences are captured, parents and children who are participating in the program or have otherwise interacted with the program will be randomly selected from communities. Given difficulties in identifying multiple participants in rural LMIC contexts, snowball sampling will be used when recruiting for focus groups after the initial two or three participants are randomly selected, utilising community knowledge to gather others with insights.

The exact sample size for each participant type cannot be determined a priori. Data collection will cease once saturation has been reached [25]. This will be assessed by the investigators in daily de-brief meetings held with the data collection team. These debriefs will be important to identify key emerging themes that require further clarification or investigation, continually build data collection capabilities and plan strategies for purposive participant selection as most relevant to the local context [26]. Variations in key findings across participant groups and geographic areas will be also be discussed during daily debriefings. Appropriate participation selection strategies that increase variation and seek clarification will be implemented. Table 3 below presents how the various components of the evaluation will be assessed with these data collection methods.

### 3.3. In-Depth Interviews

Implementing staff from CIPRB will introduce the data collectors to the community. Communities will be purposively selected to ensure that the evaluation includes sites with a variety of geographic, demographic and program characteristics. Within communities, parents and children will be randomly selected by data collectors, while program staff will be purposively selected based on their roles. The interviews will last between 30–60 min and be conducted in a quiet place such as a participant’s home or local office. Only data collectors and participants will be present. 

### 3.4. Focus Groups (FGDs)

Each FGD will comprise of 6–8 participants and will last for 40–60 min. Researchers will purposively select program staff or randomly select children and parents for FGDs and then use snowballing to identify further participants. FGDs will be homogenous and comprise of one type of participant of the same gender, such as mothers of participating children, or male CSIs only. FGDs will be held in a neutral place such as a school or community hub. 

### 3.5. Observations

There are four major components of the programs that will be observed by the research team, which are (1) review of the programs’ documented SOPs; (2) program delivery venues; (3) delivery of programs to children; and (4) supervision visits of the Anchal Maas, Anchal Assistants and CSIs.

The observations will incorporate shadowing techniques where data collectors will use prompts to ask participants their motivations behind behaviours when feasible to do so without interrupting the session [27]. 

Female data collectors will conduct observations of Anchal centre operations from 9 a.m. to 1 p.m. or SwimSafe classes from the side of the pond. They will note observations in an observational checklist, which will be developed based on program SOP requirements and monitoring outcomes such as venue suitability and maintenance, Anchal staff and CSI behaviour with children, and child responsiveness to the program. Data collectors will be present at the venue before the sessions start and will remain throughout to minimize interruption.

### 3.6. Data Collection Managmenet

The IDIs and FGDs will take a semi-structured format using interview guides that will be developed in accordance with the study objectives and field-testing. See Appendix A and Appendix B for example interview guides developed. Data collectors will take notes of main points during all IDIs and FGDs, and these will be audio recorded if participants consent. All data collectors are fluent in Bengali, and have previous training and experience in qualitative data collection in the Barishal division through previous research work with CIPRB. Data collectors are employed full time by CIPRB, and have previous relationships with the program staff. Two male data collectors and two female data collectors will be engaged. 

All IDIs and FGDs will be first transcribed in Bengali and then translated to English. 20% of translations will be checked by the data collectors for quality assurance. Transcriptions will not be returned to participants due to limitations in literacy of participants and logistical constraints in re-visiting communities and locating individuals. 

### 3.7. Consent

Free and informed written consent will be sought from all participants. All participants will be informed that the data will be used to improve the delivery of the Anchal and SwimSafe programs. The voluntary nature of the study will be emphasised. Participants will also specifically indicate that they consent to being audio recorded. Written consent for child participants will be obtained from their parents or guardian. 

## 4. Analysis

### 4.1. Process Evaluation

The primary aim of the process evaluation is to understand who benefits from the programs and under what conditions. Accordingly, the process evaluation will answer key questions about the barriers and enablers for the implementation and uptake of the programs in the Barishal division. These will be structured around context, implementation and mechanisms of impact as per the United Kingdom’s Medical Research Council guide to process evaluations [15]. These components are in line with other common frameworks for process evaluations [28,29]. The measures for these domains are described in Table 3 above. Additionally, the analysis will be used to identify bottlenecks to accessing the program and receiving its full benefits. Bottlenecks are defined as those determinants or factors that constrain coverage of the intervention. Bottleneck determinants can be classified according to UNICEF’s Monitoring Results for Equity Systems (MoRES) under four domains [30]. These domains are Enabling Environment (including social norms, policy and budget), Supply (including availability of inputs and access to services and facilities), Demand (including financial access and continuity of use) and Quality of services.

### 4.2. Gender Analysis

To identify unexpected benefits (or harms) related to gender, this process evaluation will be seeking to identify how the programs have affected the gendered roles and relationships within the communities in which they operate, including impacts for parents, program staff and other community members. Gender analysis provides insight into benefits of programs beyond the targeted effect, and can lead to the identification of possible opportunities to promote gender equity in both the program implementation team and targeted communities [20].

We will undertake a gender analysis informed by the Gender Integration Framework by FHI 360 upon which questions will be developed for IDIs and FGDs [22]. Specific questions that explore each of the five domains will be incorporated into the data collection tools. The Gender Integration Framework is specifically designed to analyse gender dynamics in development programs. This framework will allow for the analysis of the roles that men, women, boys and girls play within the program and wider community, and how these interact with power imbalances and affect opportunities, needs, constraints and relationships across the five key domains outlined in Table 4.

It is important that gender analysis be accompanied by action to address imbalances in gender power relations. We will use the findings from the gender analysis to identify how gender inequities can be addressed through practical strategies in the programs which will be implemented in mid-2019.

### 4.3. Data Analysis Methods

Quantitative data collected from program records will be analysed to determine specific process measures such as reach, fidelity and dose. Descriptive analysis such as counts and percentages will be conducted for this data to identify demographic differences between outcomes such as attendance and graduation rates. Differences in rates and proportions between groups will be analysed using bivariate Pearson Chi-squared tests and trends analysed using Chi-square linear-by-linear associations. The SPSS statistical software package will be used to conduct the analysis [31].

Analysis of the transcribed qualitative data will be assisted by NVivo 12 software [32]. A framework method of analysis will be used to generate categories and codes and will incorporate both deductive (pre-determined) and inductive (developed) analysis. This approach allows for the exploration of specific themes (e.g., barriers and facilitators of implementation) while not restricting the emergence of unanticipated themes [25,33]. As per the realist approach, the inductive process will seek to identify patterns in the qualitative data set that are reflective of the true underlying mechanisms affecting the programs’ delivery and response [18]. The quantitative and qualitative data will be drawn together to provide a contextualised understanding of who benefits from the programs and under what conditions. Qualitative analysis will be conducted independently by two separate teams. Discrepancies in findings will be discussed once independently analysis has concluded.

Quantitative and qualitative data will be presented together to illustrate program implementation status, encompassing both key successes and issues in regards to program delivery, management and community response. Quantitative data will provide insights into the current status of these implementation components, and the qualitative findings will identify the mechanisms behind these and potential strategies for improvement

All data will be stored and shared via secured servers. All files will be de-identified before sharing with translators or being used for analysis. Data will be stored for five years in the secured servers and physical cabinets for 5 years, as per Australian research ethics requirements [34].

## 5. Ethics Approval and Consent to Participate

Local ethics approval from the ethics committee of CIPRB has been granted (Memo no: CIPRB/ERC/2017/24). Ethical approval was also obtained from the University of New South Wales Human Research Ethics Committee (HC: 180608). Written informed consent will be obtained from all participants for qualitative data collection. Consent will be taken from parents or guardians of minors participating in the study.

The findings of the analysis will be presented in a report to the funder of the programs and CIPRB program staff. Findings will also be verbally shared with community VIPC members by Supervisors to gain their assistance in improving implementation. The results will additionally be published in peer-reviewed journals and presented at academic conferences.

## 6. Discussion

This protocol outlines in detail the methodology, methods, and analyses that will be used to conduct a process evaluation of the Anchal and SwimSafe drowning reduction programs in the Barishal division of Bangladesh. This process evaluation will explore the implementation of these programs through analysis of its delivery and effect on gender roles and responsibilities [20,22]. The results of this evaluation will be useful for the quality assurance and improvement of these programs. Firstly, the process evaluation components will allow the implementers to identify and remedy inadequacies in their delivery and supervision of the program, as well as better tailor program delivery to the expectations and requirements of program participants. The gender analysis will provide insights into the programs’ effect on gender roles and relationships, and allow the program to be better designed to address gender inequities.

The results may additionally guide other organisations seeking to implement community-based drowning reduction programs in LMICs by providing insights into key considerations and challenges when delivering these programs.

## 7. Strengths and Limitations

This project will engage a variety of relevant stakeholders, from participants to program delivery staff to supervisory staff, to obtain a thorough representation of the current implementation status of Anchal and Swimsafe programs. Furthermore, the incorporation of the Gender Integration Framework in the analysis allows for a more holistic view of unanticipated benefits and harms, beyond the scope of traditional process evaluations.

A key challenge for this study will be the translation of qualitative responses into English for analysis, introducing the possibility of incorrect semantics and meaning being represented in the transcripts. It is essential that key findings be compared to field notes and corroborated with Bengali-speaking data collectors to ensure no mistranslations have occurred. Data collectors will crosscheck a random selection of translated transcripts (20%) against their notes to ensure that appropriate meaning is being conveyed [35,36]. The data will be evaluated for usability once obtained to ascertain its appropriateness for analysis. Additionally, all available data from across the programs will be used in the quantitative analysis and triangulated with qualitative data to find common issues in our results. Another limitation is that the presence of data collectors at community Anchal and SwimSafe sessions may impact behaviour of community members. Hence, it will be important to cross-validate observations made with accounts from IDIs and FGDs of session operations. Lastly, we may face challenges in gathering a representative sample of participants for qualitative data collection. For example, parents with above-average negative or positive experiences of the programs may be more likely to participate. It is essential that qualitative data collection only ceases when new concepts are no longer being identified. Our sample selection is purposively guided to ensure maximal variance. In particular, daily debriefing sessions will provide an opportunity for data collection and investigation teams to collaboratively solve issues in the field in real time. It will also be imperative to prevent groupthink during debriefs by following a structured agenda where data collectors first present findings from their notes, and then discuss the implications after all points have been shared. 

## Figures and Tables

**Figure 1 ijerph-16-00968-f001:**
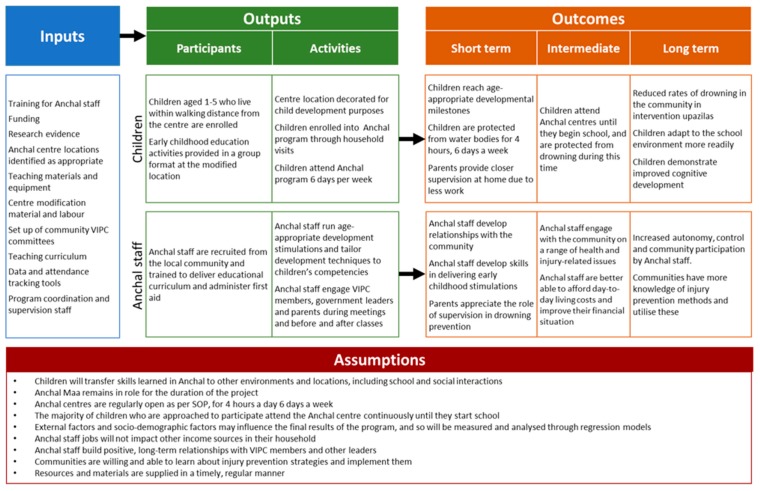
Logic model for the Anchal program.

**Figure 2 ijerph-16-00968-f002:**
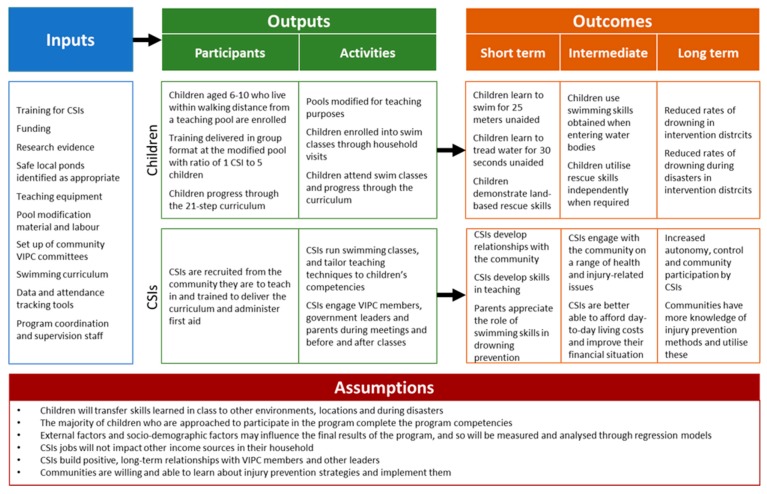
Logic model for the SwimSafe program.

**Figure 3 ijerph-16-00968-f003:**
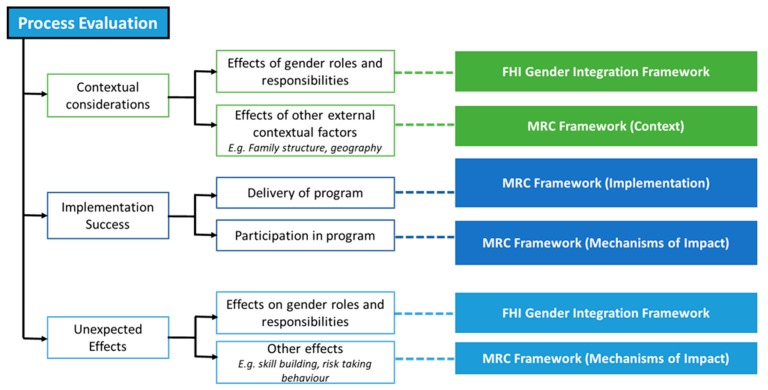
Process Evaluation methodological framework developed for this study.

**Table 1 ijerph-16-00968-t001:** Program delivery targets.

*Upazila*	Anchal Centres	SwimSafe Ponds
Kalapara	200	33
Taltoli	80	12
Betagi	120	20

**Table 2 ijerph-16-00968-t002:** Quantitative data sources for Anchal and SwimSafe.

Data Source	Variables/Indicators *
Baseline Survey	Types and rates of different reasons for enrolment or non-enrolment Child demographics in community Types and rates of different child supervision methods
Enrolment Form	Dates of commencement Demographic characteristics of child Child skills and interests
Attendance Book	Child attendance and drop out Class cancellations
Parents Meeting Minutes	Number and types of complaints Types and rates of different reasons for drop out Types and rates of different issues with access to program
Human Resources Data	Child attendance Course completions Recruitment of program staff Resignations of program staff Program staff work hours
Cluster Meeting book	Implementation costs and time Adherence to program operational guidelines Condition of program locations
Program Monitoring Forms	Instances of non-adherence to program operational guidelines Condition of program locations Adequacy of resources and equipment

* Variables are applicable to both Anchal and SwimSafe programs.

**Table 3 ijerph-16-00968-t003:** Process measures and key questions.

MRC Component	Sub-Component	Key Questions	Source of Information	Key Measures
Context	N/A	How does context shape the needs and experiences of participants and staff, and affect program implementation?	IDIs and FGDs Observations	Participant, staff, family and community experiences and perspectives
Implementation	Processes	Are support processes such as training, data collection and supply chain management sufficient to support implementation?	Program data IDIs and FGDs	Staff experiences and perspectives Efficiency of program delivery
	Reach	Do the programs access different demographic groups (ethnicity, religion gender, age) equally? Who is being missed by the programs?	Program data IDIs and FGDs	Comparison of the socio-demographic characteristics participants and non-participants Descriptions of challenges faced when accessing the programs
	Fidelity	Were the programs being implemented as intended in a consistent way across sites?	IDIs and FGDs Observations	Comparison of delivery processes across sites Descriptions of the site-wise adaptations throughout implementation
	Dose delivered	Do the programs deliver sufficient services to meet delivery targets?	Program data IDIs and FGDs Observations	Number of sites Number of operational and non-operational days per site Reasons for non-delivery (e.g., due to weather, pond condition, lack of staff etc.)
	Adaptations	How is the program adapted to different sites across the program?	IDIs and FGDs Observations	Comparison of program processes between sites
Mechanisms of impact	Participant responses and interactions	Do participants engage with the programs for continued use? Were the programs acceptable interventions at the micro, meso and macro levels?	Program data IDIs and FGDs Observations	Participant registrations, attendances, completions and drop-outs Reasons for drop-outs Comparison of socio-demographic factors for low-dose and high-dose participants Participant, staff, family and community experiences and perspectives
	Unexpected consequences	What are some unexpected benefits and issues caused by the program? Are there any unanticipated harms or dangers associated with the program?	IDIs and FGDs Observations	Participant, staff, family and community experiences and perspectives

**Table 4 ijerph-16-00968-t004:** Gender Integration Framework domains.

Domain	Sources of Information	Examples of Key Gender Relations, Barriers and Opportunities
1. Access to resources	Program Data IDIs and FGDs Observations	Education, information, services, employment, benefits, freedom of movement, transport
2. Knowledge, beliefs, perceptions	IDIs and FGDs Observations	Beliefs about capabilities, self-efficacy and confidence, acceptable behaviour and value in society, child safety and protection
3. Practices and participation	IDIs and FGDs Observations	Autonomy and time to participate both within the home and in the community, types of activities and practices
4. Legal rights and status	Program Data IDIs and FGDs	Employment contracts and rights, biases in governance and policy at program and institutional level
5. Power	IDIs and FGDs Observations	Autonomy, household financial control, control over resources, decision making within the household and in the community

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
