# Peer review of "Complexity in Implementing Community Drowning Reduction Programs in Southern Bangladesh: A Process Evaluation Protocol"

_ijerph, 2019, doi:10.3390/ijerph16060968_

Round 1
Reviewer 1 Report
International Journal of Environmental Research and Public Health – manuscript review
Title: Complexity in Implementing Community Drowning Reductions Programs in Southern Bangladesh: A process Evaluation Protocol
Thank-you for the opportunity to review this manuscript. Implementation evaluation is an important and often overlooked so this is an important contribution to the literature, however there are some issues that need to be considered by the authors and I suggest addressed prior to publication. I am also concerned that perhaps the data has already been collected, and if this is a prospective protocol? (Which is perhaps an Editorial consideration). Notwithstanding this, it is a comprehensive protocol document. At times I felt the detail in the tables and the complexity of the writing and the methods distracts the reader. My comments are mostly for consideration by the authors to help clarify and refine the focus.
Abstract – clearly written.
1. Introduction – clearly written.
Page 1, line 42: Suggest including statistics relating to the other target group in the study - children 6-10 years. Only reference to statistics in 1-4 years.
Is there an overall rate for fatal and non-fatal for children in Bangladesh that could be included here? How does this compare to the other age groups?
The literature referenced is up to date including relevant sources. However, would suggest including reference to the preventing drowning: an implementation guide which supports page 2, line 51-line 54 in the introduction.
2. Program details overall Section 2 program details is well described. See comments below
For consideration Page 3 of 14 line 105 – I query the use of the plural process evaluations? I wonder if the word evaluation can also be considered the plural?
Page 3 of 14 line 206 the sentence beginning “Such evaluations address the question …….” I believe this need to be referenced? Similarly the sentences that follows “ Gaining a clear understanding of the causal mechanisms…….” Is a reference required here?– please review
3. Materials and Methods
Theoretical approach and frameworks
Page 3 of 14 line 113. For those not familiar with Realist theory or the realist approach and its use in qualitative research could you provide a bit more context for the reader here? This is glossed over somewhat by the authors and I believe needs to be more fully described if it is the theoretical approach and how it links to inductive coding for example?. There is a lot going on in this protocol Realist Theory, Gender Integration Framework and MoRES.
Could the Logic Models Figure 2 and 3 be supplementary files? What do they add to the manuscript?
3.2.2 Qualitative Data
Page 6 of 14 line 155 could the authors clarify the comment line 160 page where they write “…the sample size will be based on expected variations across upazillas and thematic saturation” Will the interviews and focus groups be transcribed and themed to ascertain saturation as you go, or is this part of the debrief process?
The authors might like to provide some reference for saturation and thematic saturation here. See for some further reading and citation “saturation focuses on the identification of new codes or themes, and is based on the number of such codes or themes rather than the completeness of existing theoretical categories. This can be termed inductive thematic saturation. In this model, saturation appears confined to the level of analysis; its implication for data collection is at best implicit” (Saunders et al., 2018)
Might be helpful to include a structure guide or sample questions that will be used to prompt the discussion in the IDI and FGD’s.
Could Table 3 be a Supplementary file? The sample sizes you suggest are subject to change so I do not think this really adds value here. Consider removing?
Page 8 line 179 – could the authors provide a reference for shadowing techniques?
Data Collection
Clarify wording line 185 where you suggest they will be audio recorded ‘if they consent’. My query is if they do not consent are they excluded or do you take notes only? Needs clarification.
Qualitative data
The authors might like to review the criteria by Tong A, Sainsbury P, Craig J. 2007 to cite here and supports their data collection
Data analysis
Page 11 of 14 please also include the quantitative statistical package you will use here line 28 -31. Some idea of the analysis would be appropriate here as the description elsewhere is quite dense and e again some of the detail is scarce e.g. are you doing comparisons between variables using bivariate Pearson Chi-squared tests for example.
Page 11, line: 32: NVivo should be referenced - NVivo qualitative data analysis Software; QSR International Pty Ltd. Version 12, 2018.
Page 11, line 42: Australian research ethics requirements – reference required “National Statement on Ethical Conduct in Human Research 2007 (Updated 2018). The National Health and Medical Research Council, the Australian Research Council and Universities Australia. Commonwealth of Australia, Canberra”.
Page 12 of 14 line 34 – 35 a reference to support the use and description of deductive and inductive thematic analysis should be provided here. In addition to the description of why you have chosen this approach perhaps. Perhaps review Thomas DR. 2006 and the Saunders et al 2018 article again may be a help here for your qualitative analysis.
4.2 Gender analysis
How is this analysis presented? Is it all narrative or do you collect variables in the program data and conduct statistical analysis using scales and items? I am unsure how this will all be analysed.
Table 5 mentions constructs such as self-efficacy, confidence and beliefs about capabilities will these all be explored and captured in the IDIs, FGDs and observations and not specific questions to individuals? How will this be analysed?
The authors suggest they will use the gender analysis to inform strategies implemented in 2019 that is why I feel perhaps the data has already been collected?
7. Strengths and Limitations
I question why the transcripts have to be translated to English when the value of the implementation evaluation may be greater for local communities in the local language. However in terms of knowledge translation and reaching the wider audience translation is inevitable. Perhaps the triangulation of data should be introduced earlier in the paper? Can the authors consider if this should be placed in the Data analysis part of the protocol manuscript.
General comments:
Page 1, line 2: Title is quite long – could it be shorter and more succinct (e.g. by removing “complexity in”).
Page 1, line 36: Consider including Swimming lessons, Rescue skills in keywords.
References:
Saunders, B., Sim, J., Kingstone, T., Baker, S., Waterfield, J., Bartlam, B., ... & Jinks, C. (2018). Saturation in qualitative research: exploring its conceptualization and operationalization. Quality & Quantity, 52(4), 1893-1907.
Thomas DR. A general inductive approach for analyzing qualitative evaluation data. Am J Eval. 2006;27:237–246
Tong A, Sainsbury P, Craig J. Consolidated criteria for reporting qualitative research (coreq): A 32-item checklist for interviews and focus groups. Int J Qual Health Care. 2007;19:349–357

Author Response
Thank you for your comments. Please find our responses in the file attached.

Reviewer 2 Report
This is an excellent and solid research proposal. It reflects the long-term research experience of some of the-authors. As the authors point out in the limitation paragraph, the correct understanding and translation of the questionnaire needs maximum attention. Maybe some extra effort should be created (although I should not know what effort at this moment) to have this weakness strengthened as much as possible, Another issue to be considered is that the daily debriefing sessions may lead to some "groupthink" or tunnel-vision that could affect the non-verbal attitude of the data collectors or their interpretation of the replies.
Author Response

(The authors gave the same response as above.)

Reviewer 3 Report
Thank you for providing me with the opportunity to review the manuscript, “Complexity in Implementing Community Drowning Reduction Programs in Southern Bangladesh: A Process Evaluation Protocol.”
In general find this paper rather "thin." A protocol isn't all that interesting because it doesn't present results. I would be more encouraging of publishing this protocol if it were stronger. The qualitative aspects of it are weak. My feedback appears below.
Please provide supporting evidence for this claim: Two of these effective interventions within Project BHASA are Anchal and SwimSafe.
Please map these interventions onto the WHO guidelines more clearly – which initiative supports which guideline?
What is the role of the bamboo structure in the pond? Clarify.
What is the evidence base upon which the 25m of swimming, 30s of floating, and rescue techniques are based?
The remaining children “watch from outside.” Outside of what? They are already outdoors…Outside of the bamboo enclosure?
What is the gender breakdown of those in the various levels of staff? What ages are the staff members? What training do the supervisors have?
Please provide data from previous studies that show that this approach is cost effective for reducing drowning. Please also show the statistics for drowning decreases.
When you first describe the mixed methods, you write that you’ll use qualitative and quantitative data, but then start with your description of quantitative data. Please use the same order.
The overview for the qualitative data approach is weak. There is no justification given for why interviews and focus groups will be used. These approaches collect different types of data. Further, narratives are very particular forms of qualitative data. I do not think the authors actually intend to use this approach, so this term should be avoided. More information is needed concerning the focus groups. Will they be made up of only children, only parents, and only program staff, or will these FGs be mixed? The concern with bias is largely misplaced. In qualitative research, bias is not really a concern. Snowball sampling makes little sense in terms of use with random sampling, as they work at cross-purposes in many ways. The authors need to engage more deeply with qualitative research methods and approaches for this section to be convincing.
The most concerning aspect of the proposed research is the section on observations. Much, much more detail is needed. What sort of behaviours will be watched? Who will be doing the observations (male? Female? Local?)? The observations seem to be very overt and announced. To what degree are you concerned about that changing observed behaviours?
The section under the heading "data collection" should be distributed so that the info appears with each method of data collection. Again, the information on observations is incredibly sparse.
Replace the word “impacted” with “affected”
How will the gender analysis be reflected in the data collection? Please provide more information.
The qualitative analysis is unconvincing. This sounds like thematic analysis to me. Please read up on thematic analysis and use it to more clearly drive your data analysis. How will observations be coded? The idea that the two teams of analysts for qualitative data should find the same thing is problematic and hints at attempts to get at the Truth and a positivistic approach to research. Thematic analysis is constructionistic in nature. Due to one’s positionality, one would not expect to find the same results from the same data sets if analyzed by different people.
Has university ethics clearance been received? Only local is mentioned.
The discussion mentions, “ effect on gender roles and responsibility.” I’m not convinced I see how these data will be collected and analyzed.
In general, the authors need to attend much more clearly to the qualitative content of this paper and also to outline more clearly the sequencing of the data collection.
Author Response

(The authors gave the same response as above.)

Reviewer 4 Report
This is a well-written paper that describes a detailed proposal for an evaluation. The introduction is informative. There is sufficient information about the program details for the reader to appreciate the intervention programs and settings. The materials and methods have been explained well. The design of the protocol might prove useful to others involved in delivering and evaluating community-based interventions. However, I would expect to read some results of the evaluation, even if they are only preliminary results. It is not possible to assess the usefulness of the evaluation protocol without a report of the results of the evaluation and any difficulties experienced in the process.
Author Response

(The authors gave the same response as above.)

Round 2
Reviewer 1 Report
International Journal of Environmental Research and Public Health – R1
Manuscript ID ijerph-445707
Title: Complexity in Implementing Community Drowning Reductions Programs in Southern Bangladesh: A process Evaluation Protocol
The authors have diligently addressed the questions and comments I raised in my review. Thank-you for your attention to each comment. I note that you have incorporated suggested references.
General comments:
I still query the gender analysis component as the authors state the all the AMO’s are female and 3 out of 20 supervisors are female, is it gender imbalance that you are exploring?
The inclusion of the Interview guide is a good addition to the manuscript. I suggest you pilot test the guide and/or remove some of the questions that are leading and use prompts for example see section 5 below.
Anchal Interview Guide
Section 3 Perceived value
I would swap the question order in this section make Q25 become Q24
I would remove ‘or’ not from current Q 24 24. Do you think every child aged 1-5 years old should attend an Anchal Centre? or not?
Section 5 Everyday life
Q35 What are the positive benefits of being an Anchal Maa? And then you specifically ask negative – is this leading?
Should it be:
What are the benefits of being an Anchal Maa?
Probe both positive and negative?
Do you ask the Anchal Maa’s of they can swim?
I would encourage authors to do a final proof read especially of the Interview Gide (it may be the translation and native/non-native language issues however some of the sentence syntax and/or grammar needs to be actioned). Also you need to check referencing of the statistical software and how it is formatted in text.
Good luck with publishing the manuscript.
Author Response
Thank you for your comments. Please see our response as attached.

Reviewer 3 Report
My overall concerns remain the same. While the authors have addressed my previous comments, I feel that some of the responses are rather superficial and don't get to the root of the issue - in particular, I find the analysis weak, the observation section weak, and the overall content weak in terms of the contribution it can make.
Author Response

(The authors gave the same response as above.)

Reviewer 4 Report
Thank you to the authors for their response to my comments on the first version of the paper. The authors response to my comments was that "the purpose of this paper is to declare our study design and methods before data collection completes and gain feedback to improve these". As a reviewer, my feedback is that it is necessary to include some preliminary results of the evaluation and any difficulties experienced in the process. At the very least, the reader needs to be informed of any difficulties experienced when using the protocol and how these were addressed.
Author Response

(The authors gave the same response as above.)
